# Schedule and magnitude of reproductive investment under immune trade-offs explains sex differences in immunity

C. Jessica E. Metcalf [1,2] & Andrea L. Graham[1]

Sex differences in immunity are found in many species. Known immune mechanisms in birds and mammals suggest that pathogen detection may be amplified in females, whereas in males, pathogen killing is amplified. We show that these immunological profiles emerge as distinct peaks on a fitness landscape defined by sensitivity-specificity and infection-immunopathology immune tradeoffs. What selection pressures might drive males and females towards separate peaks? Surprisingly, modeling immune trade-offs alone results in a pattern of sex differences that is the reverse of what is observed. By integrating these trade-offs into a life-history framework, where the schedule and magnitude of reproductive investment differs between the sexes, we find that increased age-specific infection and mortality risks during parental investment can push females towards the peak that aligns with empirical observations. Overall, our model suggests enhanced pathogen detection (in females) versus enhanced pathogen killing (in males) is best explained if shared immune tradeoffs interact with sex-specific reproductive schedules and risks. We suggest ways to test this framework empirically.

[1] Department of Ecology and Evolutionary Biology, Princeton University, Princeton, NJ 08544, USA. [2] Woodrow Wilson School of Public and International Affairs, Princeton University, Princeton, NJ 0 8544, USA. Correspondence and requests for materials should be addressed to C.J.E.M. (email: cmetcalf@princeton.edu)

Understanding the determinants of differences in male and female health outcomes is an important goal for public health, and an increasingly recognized challenge in the study of immunity[1]. Sex differences in immune function will reflect the outcome of interacting and sometimes conflicting evolutionary processes, including selection to survive pathogens, compete for mates, and invest in offspring[2–4]. Disentangling these interacting processes is key to interpreting and understanding sex differences in immune function[3,5,6] (noting here that our focus is on sex rather than gender, which may also have its own health implications[1]).

Human females are often characterized as mounting a stronger immune response than males, a description which broadly captures females' lower incidence of many infections[7], stronger antibody responses to vaccination[8], and strikingly greater vulnerability to auto-immune diseases[1]; these characteristics are echoed in females of species across the tree of life[8] from sea urchins[9] to poultry[10]. Yet this generalization obscures important qualitative sex differences in immunity. Pathogen detection is often amplified in female mammals, with more efficient antigen-presenting cells[11], greater concentrations of B-cells[10], and greater intensity of somatic hypermutation in B-cells[12] than in males. By contrast, experimental evidence across species of birds and mammals indicates that estrogen reduces cell-mediated immune function[13], such that killer T cells are generally at lower frequencies in females relative to males, and estradiol reduces secretion of pro-inflammatory cytokines, such as TNF-α. Females also have fewer natural killer cells[8]. These differences indicate that, while pathogen detection might be more effective in females of many species, targeting and destruction of infected cells might be swifter and more intense in males, and potentially more damaging (e.g., septic shock has worse outcomes in males)[12,13].

Our aim is to investigate the ultimate drivers of these immune differences between the sexes. We hypothesize that qualitative immune differences between the sexes (beyond the quantitative differences in immune response magnitude that are often a focus (e.g., ref.[14]) emerge from the interaction between trade-offs underpinning immune function and demographic differences between the sexes. To test this idea, we model how host survival is constrained by classic trade-offs across life history stages, accounting for infection and immunity. Our analysis starts from trade-offs at the immunological scale, which emerge first from the challenge of discriminating between self and non-self, and then from the challenge of setting the appropriate scale of the immune response to control the pathogen while avoiding immunopathology. Then, to understand how optima at the immunological scale translate into sex-specific immune responses, we frame these immune trade-offs within a demographic representation encompassing sex-specific reproductive schedules and risks. We also address the particular immune challenges of pregnancy. We find that the trade-offs combine to generate two distinct optimal defense strategies, one that maximizes pathogen detection and one that maximizes response magnitude. We then investigate why male and female mammals arrive at different optima. We conclude that the female demographic schedule and costs of parental investment are key drivers of the evolution of sex differences in immune function. We finally suggest research directions to empirically evaluate and expand our results.

## Results
**Defining immunological trade-offs**. We start by developing a mathematical framing of our two focal immunological trade-offs, that of discrimination, and that of scale of response. For the first, overlap between characteristics of pathogen and host molecules means that host receptors (such as B-cell receptors or Toll-Like Receptors) run the risk of triggering inappropriate immune reactions, by responding to "self". Core principles from epidemiology capture the trade-off inherent in this discrimination problem[15]—increasing "true" positives via increased sensitivity will come at the cost of increasing "false positives", and vice versa (Fig. 1a), where the shape of the relationship is dictated by the degree of overlap between the distributions of self and non-self (see Supplementary Figure 1). Second, moving from immune detection to immune response, greater deployment of immune effectors should drive more effective pathogen control, but potentially at the cost of greater damage to the host[16–18], indicating a trade-off between the mortality hazard associated with infection, and the mortality hazard associated with immunopathology (Fig. 1b).These two trade-offs can be combined into an expression for host survival that allows us to precisely characterize what defines the level of sensitivity that maximizes survival within an age class (see "Defining survival probability in the context of a discrimination trade-off" in the Methods). At every age, individuals experience a background hazard of mortality $\mu_b$ (assumed constant over age, for simplicity), which may be increased by the mortality hazard associated with failing to detect an infection $\mu_d$ (the result of a "false negative" associated with low sensitivity, $s_e$, see Fig. 1a), or associated with immunopathology $\mu_i$ (the result of a "false positive" associated with high sensitivity, $s_e$). High sensitivity, $s_e$, is favored where the probability of infection $i_x$ at age $x$, is high, and where the hazard associated with immunopathology, $\mu_i$, is low relative to the hazard associated with infection, $\mu_d$. Conversely, low sensitivity is favored if infection probability is low and the hazard associated with immunopathology is high. Similarly, we can characterize what shapes the optimal magnitude of the immune effector response that is induced following pathogen detection (see "Extension to reflect a trade-off around the magnitude of the immune response" in Methods). The optimal magnitude of immune response can increase or decrease with probability of infection, $i_x$ and sensitivity, $s_e$ depending on the relative magnitude of other parameters (see Methods, eqn [8]), but an increase in the relative impact of immunopathology under conditions of infection (captured by the parameter $\rho$) consistently reduces the optimal level of immune related damage, $\mu_i$, and therefore the optimal magnitude of effector response.

**Evaluating fitness outcomes of immunological trade-offs**. Combining both trade-offs, a bimodal survival landscape emerges. To maximize survival, strategies with high sensitivity are paired with a low magnitude immune response (e.g., one involving few effector cells; Fig. 1c, top right) and vice versa (Fig. 1c, bottom left). Where sensitivity is high, the mortality hazards associated with infection are experienced rarely, so the survival benefits of a reduced magnitude immune response associated with reduced immunopathology (reflected by points towards the right of the trade-off on Fig. 1b) outweigh the costs of reduced response to infection (and thus high mortality hazard associated with infection, $x$ axis, Fig. 1b). Conversely, survival is maximized for strategies with low sensitivity paired with a high magnitude immune response: under this configuration, frequent infections associated with failure to detect "non-self" are best combatted with a large magnitude immune response. Increasing pathogen incidence expands the scope where high sensitivity is optimal (Supplementary Figure 2) but the bimodal landscape remains. The shape of the two trade-offs (the first, linking sensitivity and specificity, and the second, reflecting positive and negative effects of immune effector responses) will also influence these outcomes[19]. The former is to some degree constrained (illustrated and discussed in Supplementary Figure 1), and while the

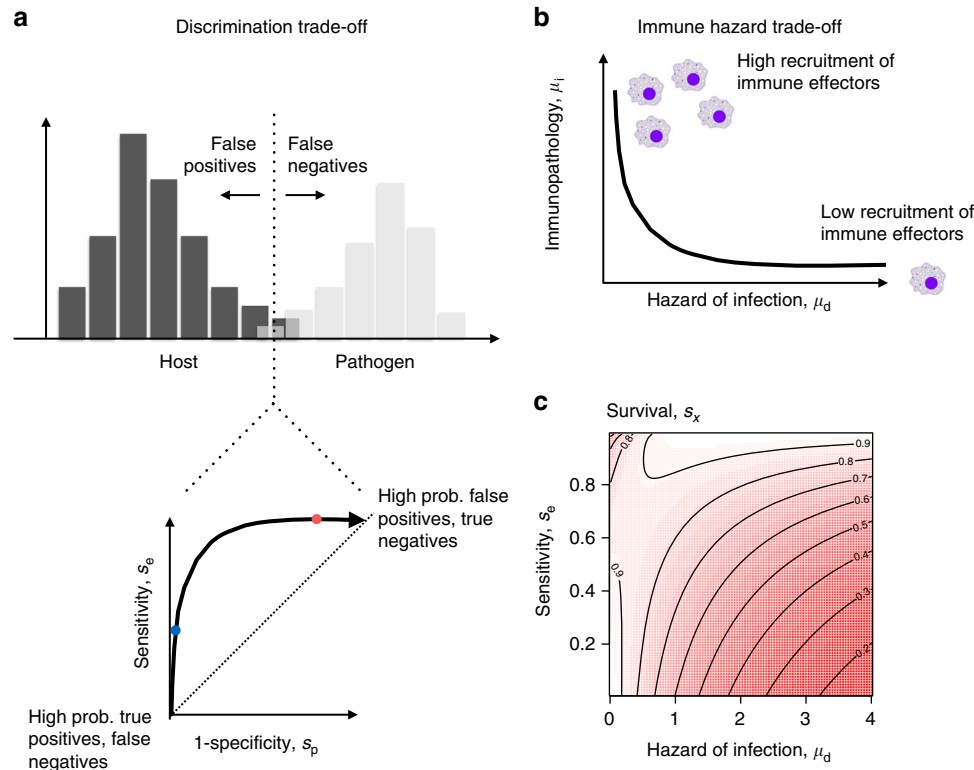

**Fig. 1** Trade-offs underpinning differences in immune function. **a** Detection and discrimination: Selective pressure on pathogens to mimic host molecules to evade detection will result in overlap between the distribution of host (black histogram) and pathogen (gray histogram) molecules, which the immune system must discriminate. Host immunity then faces a trade-off between reducing false positives (triggering a response in the absence of pathogens) and reducing false negatives (failing to detect pathogens), as delineated by a classic Receiver Operator Curve (ROC) from epidemiology obtained by plotting sensitivity (proportion of true positives detected) against 1-specificity (proportion of true negatives detected). **b** Magnitude of immune response: a high magnitude response risks considerable damage associated with immunopathology, and thus an associated hazard of mortality ($\mu_i$, y axis), but may more effectively control the pathogen, thus reducing the hazard of mortality associated with infection ($\mu_d$, x axis), note that a variety of trade-off shapes are possible. **c** Optimizing survival: Combining these two trade-offs yields a bimodal landscape of survival, with high survival (lightest shades of red) corresponding to either low pathogen detection (low sensitivity, $s_e$, y axis) and low risk of pathogen associated mortality ($\mu_d$, x axis) corresponding to high levels of immune response and thus $\mu_i$ (bottom left corner); or high pathogen detection (high sensitivity, y axis) and high risk of pathogen associated mortality ($\mu_d$, x axis) resulting from a relatively small magnitude immune response (top right corner). To obtain the surface, we set: $\gamma = 4, \eta = 0.8, \mu_b = \frac{1}{60}, i_x = 0.5, \rho = 0.01$; see Supplementary Data 1 for R code

exponential framing used for the latter simplifies derivation of the optimal magnitude of the immune effector response (above), the bimodal survival landscape emerges for a range of shapes of the relationship linking $\mu_i$ and $\mu_d$ (see Supplementary Data 1 (code) for an example with a linear relationship). Additionally, the remainder of results presented do not depend on the exact shape of the relationship (beyond it being negative). Finally, we note that trade-offs surrounding immune sensitivity and response magnitude are likely to shape whether a host resists and/or tolerates pathogens[20] but that the two survival peaks described above do not simply map onto resistance versus tolerance.

**Can these trade-offs drive sex differences in immune function?**
Although the existence of alternative optima (Fig. 1c) aligns with the observation of divergent immune strategies between the sexes, it provides no explanation as to why one sex (females in many vertebrate species) should evolve towards the high sensitivity optimum. Demographic variability across the life-span will ultimately modulate how changes in immune function within an age class translate into changes in fitness. This might shape differences between the sexes, tipping the balance in favor of one or the other optimal strategies that emerge at the within-age scale

(Fig. 1c). A classic difference between the sexes is that greater variability in male reproductive success means that males can obtain greater fitness returns for investment towards securing mating opportunities than females[21], leading to sexual selection for display or competition in males, potentially at the expense of immune function. In birds and mammals, for example, testosterone has been suggested as a mediator of allocation of resources between immune function and features relating to male–male competition, as an explanation for its broadly immunosuppressive effects[22]. Males which reduce investment in the magnitude of the immune response (e.g., reducing $R = \frac{\mu_i}{\mu_d - \mu_{id}}$, thus shifting from the top left to bottom right on Fig. 1b; see Table 1 for term definitions, and note that the shape of the relationship linking the various hazards does not enter into this expression) or in discrimination (e.g., reducing $\gamma$, the parameter governing the shape of the relationship between sensitivity and specificity, and thus increasing the perceived overlap between the distribution of self and non-self, Fig. 2, lower panel, Supplementary Figure 1), might experience a fertility advantage. Females would experience no such advantage, since whatever the resources invested, the number of offspring that females can produce within a time horizon has an upper bound. However, reducing allocation towards $R$ always results in selection for increased sensitivity all

**Table 1 Model parameters, description, and, where appropriate, defining relationships characterizing the core trade-offs under investigation**

| Symbol | Description | Defining relationships |
|---|---|---|
| $\mu_b$ | Baseline mortality hazard | |
| $\mu_d$ | Mortality hazard reflecting undetected infections | |
| $\mu_i$ | Mortality hazard resulting from immune activity in the absence of infection | $\mu_i = \exp(-\eta\mu_d)$ [used for analytical solutions for $\mu_d^*$only, see Results section entitled "Evaluating fitness outcomes of immunological trade-offs"] |
| $\mu_{id}$ | Mortality hazard resulting from immune activity in the presence of infection | $\mu_{id} = \rho\mu_i$ [used for analytical solutions for $\mu_d^*$only, see Results section entitled "Evaluating fitness outcomes of immunological trade-offs"] |
| $\eta$ | Parameter controlling the trade-off between mortality hazard associated with undetected infections, and mortality hazard from immune activity | |
| $\rho$ | Parameter controlling the magnitude of mortality hazard associated with immune activity during undetected infections | |
| $s_e$ | Sensitivity, or probability of detecting a "true positive" or infection presence | $s_e = 1 - exp^{-\gamma(1-s_p)}$ |
| $s_p$ | Specificity, or probability of detecting a "true negative" or infection absence | |
| $\gamma$ | Parameter controlling the sensitivity/specificity trade-off. | [See methods, Supplementary Figure 1, lower panel Fig. 2] |
| $s_x$ | Survival at age $x$ | [See methods] |
| $i_x$ | Probability of being infected at age $x$ | |
| $\lambda$ | Population rate of increase | |
| $w_x$ | Stable population structure for individuals of age $x$ | |
| $v_x$ | Reproductive value for individuals of age $x$ | |

else being equal (see Methods, eqn [4]), a prediction at odds with the observation of higher sensitivity in females. The effect of a reduction in discrimination depends on the context: reduced discrimination selects for increased sensitivity in contexts with prevalent and damaging infection (Fig. 2, top left, the expected direction), but selects for higher specificity in contexts where infection is rare and immunopathology is damaging (Fig. 2, bottom right). While translating the range of values of $R$ and $i_x$ to reality is not straightforward, the unexpected outcome (selection for greater specificity) characterizes a large part of reasonable parameter space (Fig. 2). Overall, these two patterns (increased sensitivity with reduced $R$, and increased sensitivity across a broad scope of reduced discrimination) both suggest that allocation trade-offs between immune function and fertility alone are unlikely to drive the emergence of observed sex differences in birds and mammals (i.e., generally more sensitive females). We therefore turn to the impact of broader sex differences in the schedule and magnitude of reproductive investment.

**The impact of sex-specific reproductive risks and schedules**. In the sex making a greater reproductive investment, their mortality, infection risk, or mortality associated with infection might all be increased relative to identically-aged individuals of the other sex. Such changes will shift both the stable population structure and reproductive value of individuals across ages, altering the impact of underlying parameters such as immune sensitivity on the population growth rate, a proxy for fitness (see section "Defining the demographic context" in Methods). Addressing this requires moving beyond identifying the strategy that optimizes, $s_x$, survival within an age class (the focus so far) to identifying the strategy that maximizes fitness for each of the sexes, i.e., requiring a full demographic model. We assume that neither sex is limiting, and therefore evaluate outcomes for the two sexes separately (see Methods). Taking as a starting point constant background mortality over age, and constant fertility from the age of maturity, we can characterize the consequences of a reduction in survival during reproductive years (achieved by increasing the background hazard of mortality $\mu_b$ during those years), potentially

reflecting costs associated with parental investment (Fig. 3a) and delineate the resulting changes in fitness associated with a change in survival (Fig. 3b–d).

A higher burden of infection or mortality associated with infection while reproducing provides a straightforward way of weighting the optimal life history towards sensitivity in the sex making the greater investment (Fig. 3e–h). More subtly, changes in baseline patterns of mortality or fertility over age can modulate the optimal sensitivity, as long as there is variability in the risk of infection over age. Although baseline mortality, $\mu_b$, does not enter into the expression of the optimal sensitivity within an age class (see Methods, eqn [4]), and thus has no effect if infection risk is constant (Fig. 3e), where the risk of infection varies over age, changes in the stable age distribution emerging from changes in the baseline mortality over age will shift the contribution of different age classes to fitness (Supplementary Figure 3), ultimately altering the optimal level of sensitivity. An early age of infection favors higher sensitivity (Supplementary Figure 4), echoing basic theory on the evolution of senescence—improvements in survival earlier in life are generally favored over later ones—and this increase in sensitivity is amplified if there is higher mortality during reproductive years, since the penalty associated with years of high sensitivity and thus "false positives" is reduced as a result of reductions in life expectancy (Supplementary Figure 4). The pattern of fertility over age modulates this outcome: a higher reproductive value of later age individuals makes false positives during years unaffected by infection costly again, and thus tips the balance away from sensitivity.

**The impact of pregnancy and antibody transfer**. In mammals, the physiology of pregnancy poses a particular challenge in the context of the evolution of female immunity. Greater discrimination between self and non-self might allow more nuanced tolerance of (self-similar) offspring, but the scope of contexts where an increase in discrimination favors increased sensitivity (mapping onto observations for mammalian females) is limited (Fig. 2). Sensitive immune responses might also lead to miscarriage where fetuses are detected as "non-self", also tending to

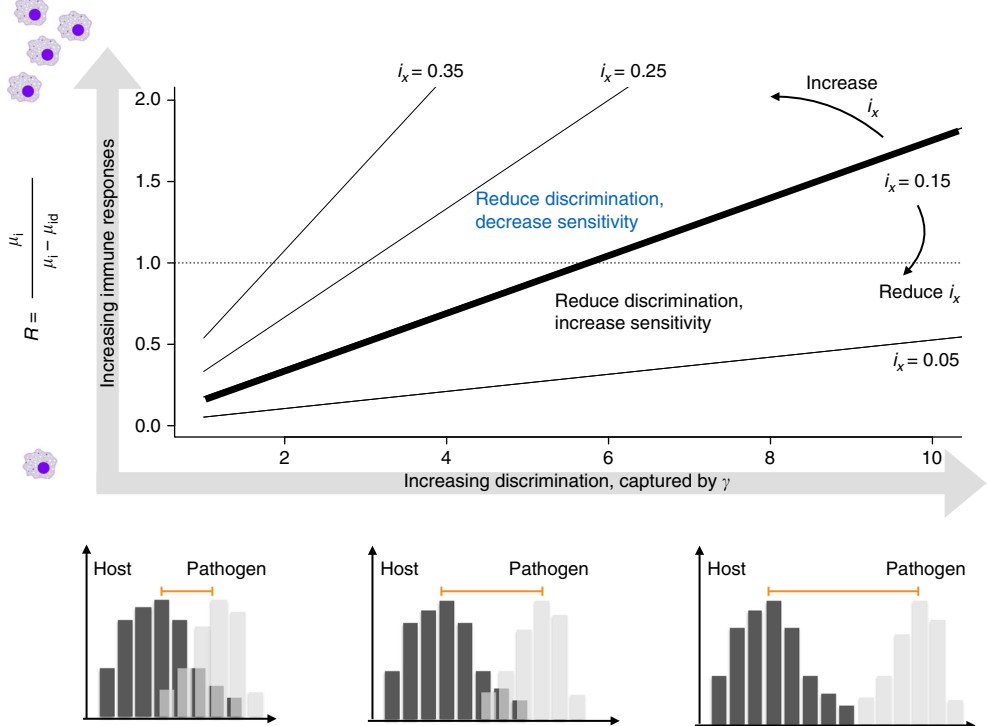

**Fig. 2** Impact of trade-offs with other physiological functions. Across a gradient of discrimination, $\gamma$ (x axis, illustrated below by a schematic of the impact on host perception of the difference between pathogen and host, see Supplementary Figure 1) and the ratio between hazard experienced in the absence and presence of infection (y axis, $R = \mu_i/(\mu_d - \mu_{id})$, where larger values indicate larger magnitude of immune responses), we can identify the scope of contexts where reducing resource allocation towards discrimination, captured by $\gamma$, reduces the optimal sensitivity, in line with expectation and observation based on male mammals. The solid lines divide the plane into areas where reducing discrimination reduces sensitivity (above the line) and those where it increases (below the line) for different levels of incidence of infection (text), illustrated for $i_x = 0.15$ (thick line), where the blue text indicates the area (relative to this value of incidence) where a reduction in discrimination leads to evolution of reduced sensitivity; and black text indicates the opposite. Sufficiently low discrimination (x axis, left) results in a large array of magnitudes of the immune response (y axis, $R$) where less sensitive immune function can evolve in response to a further reduction in discrimination (there is a large area above the thick line). If discrimination is higher (x axis, right), then the magnitude of the immune response ($R$, y axis) must be high for a reduction in discrimination to result in decline in sensitivity, and thus for predictions from the model to map onto observations for male mammals. Reducing incidence increases this span (curved arrows, for the example with $i_x = 0.05$ most the plane is above the line, so that in most contexts we expect allocation away from discrimination to reduce in less sensitivity)

drive selection for lower sensitivity in females (Fig. 3h). However, the time-scale of pregnancy being short relative to the life-span in many organisms, this might rather drive the evolution of plasticity in immune response, tilted towards reduced sensitivity during pregnancy, rather than a lifelong reduction, in line with what is observed[23].

Transfer of maternal antibodies (or, more rarely, paternal antibodies[24]) can importantly shape offspring survival[25,26], and is another potential driver of sex differences in immunity. Immune memory (so far ignored, but a pre-requisite for maternal immunity) will reduce the impact of immunopathology for previously observed pathogens. Offspring benefiting from maternal immunity will also experience a reduction in the effect of pathogens on their survival. Combining these two pieces conceptually (Fig. 4) suggests that to optimize the joint survival of mothers and their offspring, maternal immunity might reduce or increase sensitivity relative to males, depending on the relative magnitude of gains in terms of the hazard associated with immunopathology, $\mu_i$, and the hazard associated with infection, $\mu_d$. However, offspring generally have little immune function during early life, implying a negligible impact of immunopathology, and large benefits in terms of pathogen survival resulting from transfer of maternal immunity. Increasing sensitivity (shifting the discrimination threshold left) will thus incur little

cost to offspring and potentially considerable pathogen survival benefits. These benefits are likely to be sufficient to offset increases in immunopathology hazards in mothers from increased sensitivity, since early life events have magnified importance for fitness. Under these conditions, sensitivity in females (or more generally the immunity-transferring sex) will increase (see Supplementary Note 1 for details of a simulation to support the conceptual framing, plus accompanying code (Supplementary Data 1), Supplementary Figure 5 for maternal immunity results, Supplementary Figures 6, 7 for determinants of the optima).

## Discussion

To map out how sex differences in immunity could evolve, we combined trade-offs including risks and benefits associated with sensitivity and magnitude of immune response through to survival costs of parental investment. This integration points to likely drivers of the evolution of higher sensitivity in females of many species, including humans, despite the costs of greater immunopathology. Interestingly, we show that classic trade-offs associated with variance in male reproductive success cannot consistently explain lower male sensitivity frequently observed. More subtle impacts of the female demographic schedule and costs of parental investment must be considered to fully account for differences in male/female immune function. Our

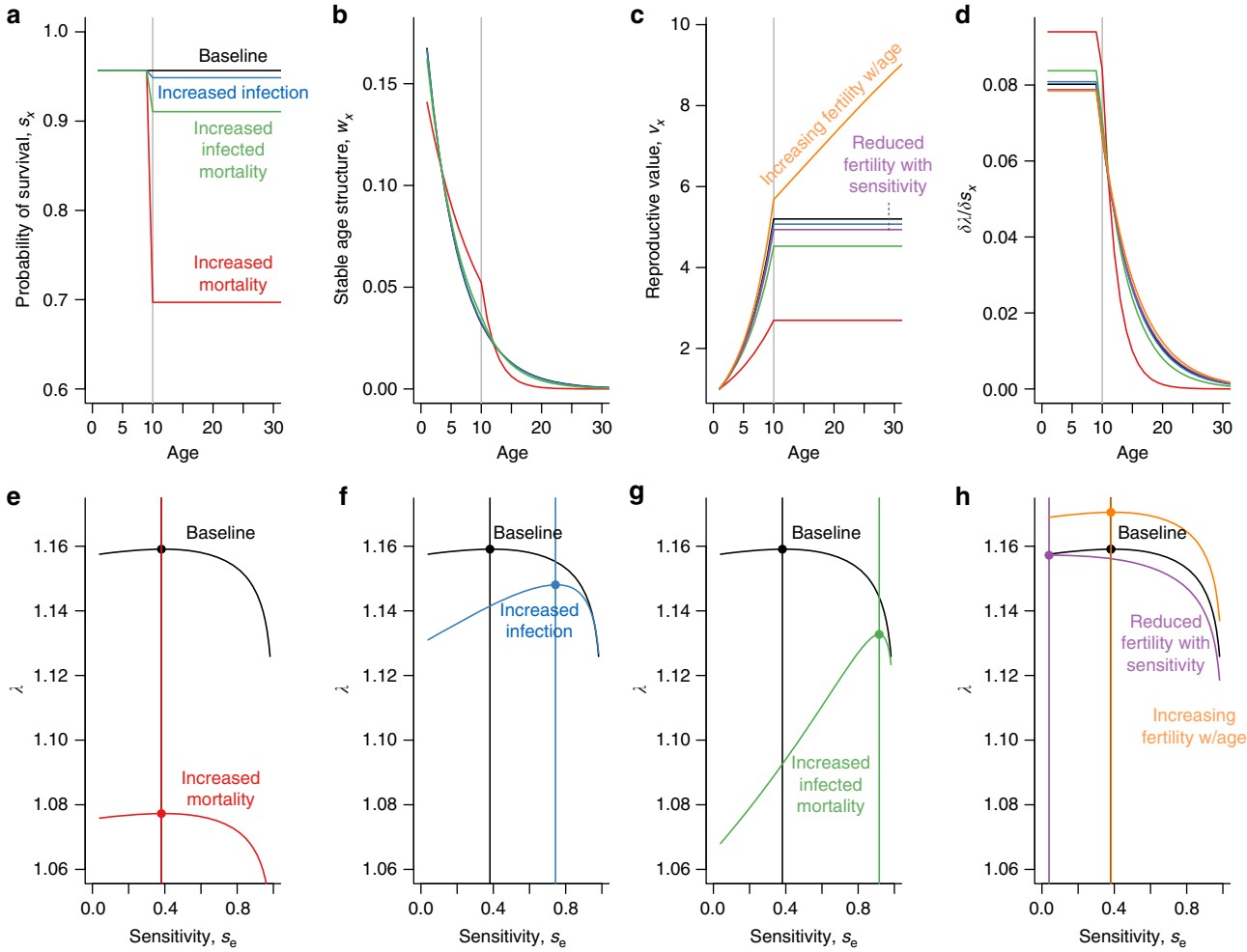

**Fig. 3** Tracking the impact of changing sensitivity from survival to fitness **a** Survival: across age (x axis) scenarios include constant survival (y axis, black), reduced survival associated with increased infection during reproductive years (blue, starting at the age of maturity indicated by the gray vertical line), reduced survival associated with infection during reproductive years, although probability of infection remains constant (green, as before), and overall reduced survival (e.g., associated with risks of birth, or other investments for in reproduction), again with constant probability of infection (red, as before); **b** Stable population structure: changes in the age profile of survival directly alter the stable population structure (colors as before); high mortality during reproductive years reduces the fraction of later aged individuals (all lines are predominantly below the black line for ages above the age of maturity) and if sufficiently large, can increase the fraction of younger aged individuals across some ages (red line is above the other lines). **c** Reproductive value: reducing survival reduces the reproductive value of individuals, colors as before with two additional scenarios added for comparison: increased fertility with age (orange), and reduced fertility for higher levels of sensitivity (purple). **d** Effect of changes of survival on fitness as measured by the populations rate of increase, $\lambda$, is the product of the stable population structure (**b**) and reproductive value (**c**); largest impacts are found for increasing survival at young ages for the scenario with the lowest overall survival. The overall optimal strategy for sensitivity (**e**) does not change with baseline mortality (colors as above, see Methods) as long as infection is constant over age (but see Fig. 4), increases with infection during reproductive years (**f**), or mortality associated with infection during reproductive years (**g**, **h**) is reduced by a reduction of fertility by sensitivity (e.g., if miscarriage is a risk during pregnancy) and unaffected by a change of fertility with age (but see Fig. 4)

analysis opens the way to testable predictions for sex differences in immune function across a landscape of parental investment (Fig. 5). Building a comparative framework to titrate how the co-evolution of parental investment and sexual selection[27] shape evolution of immune function by modulating sex differences in demographic trajectories, in the context of proximate drivers of sex differences in demography and immunity (from sex chromosomes[28] to hormones) is an important next step. Strengthening the empirical knowledge-base of sex differences for a broader array of species across the tree of life will be essential to evaluating emergent predictions. For example, although immune defenses of invertebrates, including immune memory and maternal immunity (e.g.[29–31]), are increasingly

well understood, proximate drivers analogous to likely mediators of sex effects in birds and mammals (such as testosterone[22]) remain largely unknown. Furthermore, elucidation of plasticity in sex differences[14] and the role of genetic constraints on sexual dimorphism on shaping life history outcomes[32] are important directions both theoretically and empirically. Finally, encompassing realistic age-schedules of survival (including high early and late mortality) and fertility (beyond the constant rates considered here) for which there is increasing information[33], as well as more formally accounting for the degree to which one of the two sexes may be limiting (e.g., by including a "marriage function"[34]) may generate more nuanced system-specific predictions. An improved understanding of the

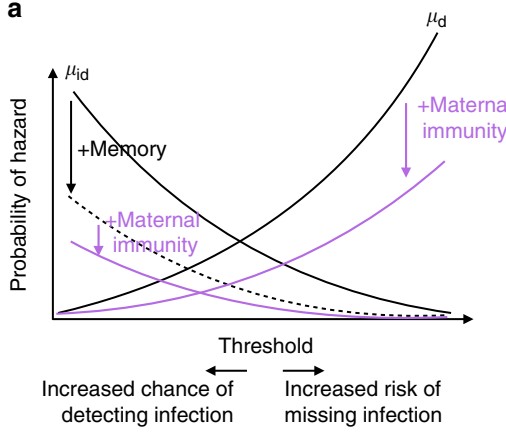

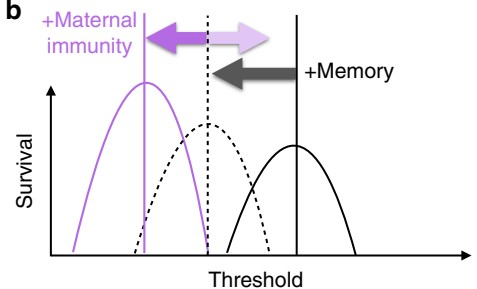

**Fig. 4** Conceptual map of the impact of memory and maternal immunity on sensitivity threshold. **a** The threshold defining the discrimination trade-off (*x* axis, high sensitivity to the left, and specificity to the right) affects the probability that an individual experiences different mortality hazards (*y* axis). Assuming that the pathogen trait that the host is using for discrimination occurs evenly across the *x* axis, the probability of infectious disease related mortality ($\mu_d$, increasing black line) increases with the threshold used to distinguish "self" (left) from "non-self" (right), because more infections are missed. Conversely, the probability of hazards arising during an appropriate immune response ($\mu_{id}$ declining black line) increases as the threshold declines, since more pathogens are detected. Memory (black arrows) reduces the probability of $\mu_{id}$, if previously seen pathogens elicit reduced self-destructive immunological activity (dashed black line); maternal immunity (purple arrows) both protects offspring from pathogen related disease in their first year of life, thus reducing the probability of $\mu_d$ (increasing purple line), and might spare offspring effects associated with $\mu_{id}$ (decreasing purple line). **b** The threshold that maximises survival (*y* axis) will reflect the minimum of the sum of $\mu_d$ and $\mu_{id}$, accounting also for immunopathology $\mu_i$ and the balance across different life stages (see Supplementary Figure 5). While memory should reduce the optimal threshold, the impact of maternal immunity will depend on the relative magnitude of the two hazards (likewise, here illustrating a decrease, but with the faded purple arrow to indicate that an increase is also possible)

ultimate determinants of sex differences in immunity will provide an alternative lens onto understanding sex differences in health outcomes, opening the way to novel interpretation of existing results, but also pointing to new research directions in this important area.

## Methods

**Defining survival in the context of a discrimination trade-off.** Individuals can be either infected, or not, and either detect (or suspect) this occurrence, or not; each outcome is associated with a particular risk of mortality, see Table 2. The probability of detection is predicated on a trade-off between sensitivity, $s_e$ and specificity, $s_p$, where sensitivity is defined as a function of specificity according to:

$$s_e = 1 - \exp^{-\gamma(1-s_p)} \quad (1)$$

such that $\gamma$ determines the discriminatory power associated with that particular trade-off, see Fig. 1. From this, we can define the probability of survival for individuals of age *x* till the next time-step as the sum of the different mortality hazards:

$$s_x = \exp\left(-\left[\mu_b + (1-i_x)\mu_i(1-s_p) + i_x\mu_d \exp^{-\gamma(1-s_p)} + i_x\mu_{id}\left(1-\exp^{-\gamma(1-s_p)}\right)\right]\right) \quad (2)$$

where $i_x$ is the probability of being infected for an individual of age *x*, $\mu_b$ is the baseline hazard of mortality, $\mu_i$ is the mortality hazard associated with immunopathology occurring in a context of a false positive $(1-s_p)$, $\mu_d$ is the mortality hazard associated with undetected infection, reflecting a false negative $(1-s_e) = \exp^{-\gamma(1-s_p)}$, and $\mu_{id}$ reflects the damage from immune response occurring in the context of true positives, $s_e = 1 - \exp^{-\gamma(1-s_p)}$, see also Table 2. From this expression, we can extract the value of specificity that yields the maximum survival for individuals of age *x*, $s_p^*$, by identifying the value of specificity for which the derivative of the expression above, $d_x/ds_p$, is equal to zero (verifying also that this is a maximum and not a minimum). The derivative relative to $s_p$ is defined by

$$\frac{ds_x}{ds_p} = s_x\left[(1-i_x)\mu_i - \gamma i_x\mu_d \exp^{-\gamma(1-s_p)} + \gamma i_x\mu_{id}\exp^{-\gamma(1-s_p)}\right] \quad (3)$$

From this, the optimal specificity is:

$$s_p^* = \frac{1}{\gamma}\left[\log\left(\frac{\mu_i}{\mu_d - \mu_{id}}\right) + \log\left(\frac{1-i_x}{i_x}\right) + \log\frac{1}{\gamma}\right] + 1 \quad (4)$$

Further, introducing $s_p^*$ into the expression for the second derivatives of survival at the optimal, $\frac{d^2 s_x}{d^2 s_p}$ yields a negative value, confirming that this value reflects a maximum. It is straightforward to relate to this quantity to optimal sensitivity, $s_e^*$, results for which are discussed in the Results section entitled "Evaluating fitness outcomes of immunological trade-offs" (analytical results are framed in terms of specificity for consistency with previous work[15]).

**A trade-off around the magnitude of the immune response.** A stronger immune response (i.e., one with a greater inflammatory response, production of natural killer cells, cytotoxic T cells, and other immune effectors) is likely to be both more damaging and more effective at controlling pathogen incursions. We can capture this trade-off by introducing a dependence between the hazard associated with undetected infection, $\mu_d$, and the hazards associated with immunopathology $\mu_i$, such that in the context of "false positives", the hazard can be defined by:

$$\mu_i = \exp[-\eta\mu_d] \quad (5)$$

and we assume that in the context of "true positives" the damage is a fraction $\rho$ of this:

$$\mu_{id} = \rho \exp[-\eta\mu_d] \quad (6)$$

As above, we can identify the value of $\mu_d$ that maximizes survival at age *x*, by taking the derivative, but now relative to $\mu_d$:

$$\frac{ds_x}{d\mu_d} = s_x\left[\eta(1-i_x)(1-s_p)\exp^{-\eta\mu_d} - i_x\exp^{-\gamma(1-s_p)} + \eta i_x\rho\left(1-\exp^{-\gamma(1-s_p)}\right)\exp^{-\eta\mu_d}\right] \quad (7)$$

From this, the optimal level of hazard for undetected infections is:

$$\mu_d^* = 1/\eta\left[\log\eta + \log\left[(1-i_x)(1-s_p) + i_x\rho\left(1-\exp^{-\gamma(1-s_p)}\right)\right] - \log\left(i_x\exp^{-\gamma(1-s_p)}\right)\right] \quad (8)$$

which can be directly related to the optimal hazard associated with immunopathology, $\mu_i^*$, which scales with the magnitude of the immune response deployed (see Results section entitled "Evaluating fitness outcomes of immunological trade-offs"); again, second derivatives are negative indicating that this corresponds to a maximum. Note that the two trade-off relationships defined above for $\mu_i$ and $\mu_{id}$ as a function of $\mu_d$ are only used to obtain analytical results for the optimal investment in immune effectors, and do not influence the broader set of qualitative results, see Results section entitled "Evaluating fitness outcomes of immunological trade-offs", Supplementary Data 1 (code).

**Defining the demographic context.** Using the expression for survival at age *x*, $s_x$ that is the result of the summed competing hazards at that age, we identified the strategy that maximized survival at age *x* in terms of both specificity, $s_p^*$, and the magnitude of the immune response, $\mu_d^*$. However, the optimal strategy will reflect

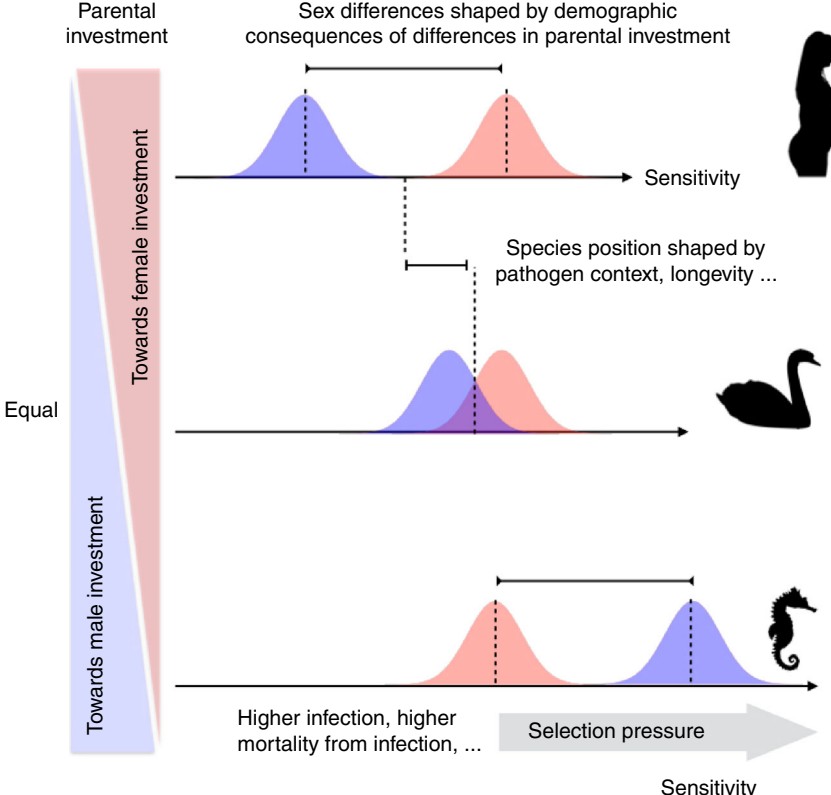

**Fig. 5** Predicted relationships between parental investment and the evolution of sensitivity: increased parental investment of females (moving from center up) or males (moving from center down) will amplify features that select for increased sensitivity (x axis), such as increased infection during reproductive years, increased mortality during infection, or increased mortality during reproductive years coupled with high early infection. In species where both sexes make similar investments (e.g., swans, middle) immune sensitivity should be similar (overlapping distributions, center); for species where females make larger investments in parenting (e.g., mammals like humans, top) sensitivity should be higher in females, and conversely for species where the reverse is true (e.g., seahorses, bottom). Co-evolution between parental investment and sexual selection will shape evolution of features that further modulate this: in mating systems where older males dominate reproduction (polygyny), male sensitivity should be further reduced relative to females given concentration of infection at young ages. These predictions are, in principle, testable across such life history diversity: sensitivity could be estimated in terms of the pathogen dose-dependence of immune cell activation (in vitro if not in vivo), where cells of relatively sensitive hosts should activate at lower pathogen doses than cells of relatively insensitive hosts

---

### Table 2 Possible mortality hazard scenarios for an individual of age x

|  | Respond | Don't respond |
|---|---|---|
| Infected ($i_x$) | $\mu_{id} \times s_e$ ("true positive") | $\mu_d \times (1 - s_e)$ ("false negative") |
| Not infected ($1 - i_x$) | $\mu_i \times (1 - s_p)$ ("false positive") | 0 ("true negative") |

Mortality hazards occur according to the probability of being infected at age x, $i_x$, combined with the probabilities defined by the discrimination trade-off (sensitivity/ specificity or $s_e/s_p$) outcome. The parameter $\mu_i$ captures the mortality hazard associated with exuberant immune reactions in the absence of infection (false positives), the parameter $\mu_{id}$ captures the same but in the presence of infection, and $\mu_d$ captures the mortality hazard associated with undetected infections (false negatives)

---

the optimal across the life-span and not within a single age class. Changes in the baseline mortality hazard or probability of infection across age, combined with the pattern of fertility over age will modulate the overall optimal strategy. Such patterns are likely to differ between the sexes, and therefore play a role in shaping the evolution of different immunological processes between the sexes.

To address this, we frame the full demographic process using a Leslie matrix[35], denoted **M**. We assume that survival from age x to x + 1, reflecting the $x^{th}$ column and $x+1^{th}$ row in the matrix, is defined by $s_x$ (such that the proportion of individuals surviving to age A will be $s_1s_2s_3 \ldots \times s_{A-1}$); and the first row of the matrix is completed to reflect a chosen fertility schedule, $f_x$. The dominant eigenvalue of this Leslie matrix, $\lambda$, captures the population rate of increase and can be used as a proxy for fitness. This framing provides a straightforward expression of the impact of changes in underlying parameters on the population rate of increase (confusingly called "sensitivity"):

$$\frac{\delta\lambda}{\delta s_x} = (\bar{v}_x w_{x-1})$$ (9)

where $\bar{v}_x$ is the reproductive value of individuals of age x and $w_{x-1}$ is the fraction of individuals in the x-$1^{th}$ age class, i.e., is the x-$1^{th}$ value of the stable age distribution[34]. For this expression, the scalar product of the vector of reproductive values (**v**, the left eigenvector corresponding to the dominant eigenvalue of **M**) and the vector reflecting the stable size distribution (**w**, the right eigenvector corresponding to the dominant eigenvalue of **M**) is scaled such that $\prec \mathbf{w}, \mathbf{v} \succ = 1$.

By the chain rule, we can obtain the dependence to underlying parameters, such as specificity, $s_p$:

$$\frac{\delta\lambda}{\delta s_p} = \sum_x \frac{\delta\lambda}{\delta s_x}\frac{\delta s_x}{\delta s_p}$$ (10)

We can thus modulate the patterns of $i_x$, $\mu_{bx}$, and $f_x$ and evaluate the outcome for the population rate of increase. For comparison between the sexes, this is equivalent to making the assumption that the mate of the opposite sex is never limiting (i.e., we are not formally modeling a "marriage function").

**Code availability**. All code used to generate results available in Supplementary Data 1 and Supplementary Data 2

## Data availability

Data sharing is not applicable to this article as no datasets were generated or analyzed during the current study.

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

## Author contributions

C.J.E.M. and A.L.G. came up with the idea; C.J.E.M. developed the code and analyses; C.J.E.M. and A.L.G. wrote the paper.

## Additional information

**Competing interests:** The authors declare no competing interests.

