## [Peer Review File · Nature Communications]

Reviewers' comments:

Reviewer #1 (Remarks to the Author):

This paper describes an extremely interesting model incorporating both sexual selection and demography in explaining sex differences in immune response. The idea of hazards and sensitivity is a very appropriate one to this question, and one that's quite novel. I liked the approach very much, but think that the generality of the work is overstated, and that it is applicable only to placental mammals, a point that needs to be made much more clearly. Furthermore, it isn't new to link life history and immunity. There is nothing wrong with these limitations, but they need to be emphasized from the start. The idea of parental immunity playing a key role is likewise new and interesting, though again I think it's limited to mammals and birds. I lack the mathematical expertise to evaluate the model itself, so leave that to other reviewers; I focus on the way that the results are interpreted and the conclusions are made.

Starting with the title, the authors really don't mean "while parenting," since this implies the difference is due to something happening during parental care, which most species don't exhibit. Similarly, later on, when they discuss "In the sex making a greater investment in parental care, mortality, infection risk, or mortality associated with infection, might all be increased relative to identically-aged individuals of the other sex," they don't really mean parental care, they mean greater parental investment. Parental investment is seen in most if not all sexually-reproducing organisms, including plants, but parental care is only found in a tiny fraction. The distinction is important, because of course even in mammals, where females exhibit far more parental care than males, males still may have substantial parental investment in the form of, for example, expensive courtship displays. How this factors into the models is an interesting question, because parental investment by males may occur for a shorter but more intense period during the life history, say during mate attraction, but can still be quite high. So the contrast isn't between high and low parental investment, it's the schedule of when those investments are made.

And in the first sentence of the abstract: "There are clear sex differences in immunology in every species for which this has been investigated," again this seems like an over-generalization. Part of why the study of sex differences in immunity (not immunology, as the authors state) has been so complicated is that sometimes the sex differences are found and sometimes they are not – several reviews note the contradictions, particularly when non-mammals are included. I realize that the short format for Nature Communications requires brevity, but the authors need to be more careful about generalizing.

Along similar lines, non-mammals in general and invertebrates in particular are very poorly studied in this field. This doesn't make the study less interesting, but it means that we simply don't know whether non-mammals exhibit similar trends. In particular, one study in pipefish doesn't make it the case that "Research on species for which sex roles are reversed (e.g., pipe-fish, where males carry offspring) is broadly consistent with this characterization." Are there studies of, say, phalaropes or spotted sandpipers or even marmosets (in which males provide a great deal of parental care) that substantiate this claim? Even calling sex roles "reversed" has been questioned, since it's not clear that when

a greater variety of species is taken into account, we can see clear roles for males and females across the board.

How is the idea that “qualitative immune differences between the sexes emerge from the interaction between trade-offs underpinning immune function and demographic differences between the sexes” different from other ideas that have been proposed? It looks to me like the demographic stuff is what’s key here, since lots of other studies point to tradeoffs, even between immunity and other life history traits (see some of the references below).

The idea that testosterone is or isn’t key to the tradeoff is interesting, since of course (related to the above points) most animals on earth don’t have it. If invertebrates show the same pattern of generally higher female immunity, what would this say about how this pattern evolved? It seems to me that one can argue that testosterone’s effect on the immune system may be an incidental mechanism, but that differential selection on males and females would produce sex differences in immunity regardless. Alternatively, testosterone’s influence could be a necessary precursor for the kind of sex differences we see, because in animals that lack such hormones, no other pathways make the link between immunity and sexual selection possible. We just don’t know, though a number of investigators have tried to examine sex differences in immunity in invertebrates. The bottom line, though, is whether this paper and its findings are just about mammals or perhaps vertebrates, and it’s hard to see from the manuscript what the authors think.

A strength of the paper is its look at different life history stages, not just assuming that all adults have the same tradeoffs between sensitivity and other traits. I also think that the literature on tolerance vs. resistance is relevant to the authors’ argument, because the idea is the same as their sensitivity/false positive vs. false negative approach; there, the tradeoff is between ramping up one’s immunity to resist a pathogen, which is costly, or not do so and simply deal with (tolerate) the effects, which may be less costly in the long run. Lars Raberg has done some of this work.

I understand that the authors are limited in their citations, but here are a few very relevant papers that would be of interest:

Zuk, M. and K.A. McKean. 1996. Sex differences in parasite infections: patterns and processes. *Int. J. Parasitol.* 26:1009-1024.

Rolff J, Armitage SAO & Coltman DW (2005) Genetic constraints, sexual dimorphism and immune defence. *Evolution*, 59, 1844-1850.

Rolff J (2002) Bateman’s principle and immunity. *Proceedings of the Royal Society of London B*, 269, 867-872.

McKean, K.A. and Nunney, L. 2005. Bateman's principle and immunity: phenotypically plastic reproductive strategies predict changes in immunological sex differences. *Evolution* 59(7): 1510-1517.

Moore, S.L. and Wilson, K. 2002. Parasites as a viability cost of sexual selection in natural populations of mammals. *Science* 297: 2015-2018.

Zuk, M. and Stoehr, A.M. 2002. Immune defense and life history. *The American Naturalist* 160: s9-s22.

Pursall, E.R. and Rolff, J. 2012. Immunopathology in ecological immunology. In:

Ecoimmunology, G.E. Demas and R.J. Nelson, eds. Oxford University Press.

Reviewer #2 (Remarks to the Author):

Please see attached file.

Reviewer #3 (Remarks to the Author):

Metcalfe and Graham examine theoretically the important question of why males and females typically differ in immune function. A real advance here is the mechanistic detail at which the authors address this question. Typically, previous studies have been much more superficial when addressing this question with a simple characterization that males have reduced immune function due to immunosuppression through testosterone which is used as a hypothesis to explain male biased parasitism alongside higher exposure in males due to behavioral differences. What is very novel here is that it is underpinned by a much more detailed understanding of sex differences in immune function: with females better at detection and males better at the response. Cutting edge life-history theory is used to determine how we can understand these differences from first principles. Well empirically underpinned trade-offs are assumed that convincingly show that these bimodal life-histories can be easily predicted to coexist. Then they examine what factors can lead to males and females being selected for less often stronger and more common weaker responses respectively. What I really like is that they show that the classic Bateman's principle arguments do not work, but that the pattern can be explained by differential mortality due to parental care. This gives testable predictions and changes the way the field thinks about sex differences in immune function. I would expect it to lead to shift in the field and be exceptionally well cited.

The paper is very well written and authoritative. It clearly benefits from this unique collaboration between two evolutionary biologists with complimentary expertise in immunology and life-history/demographic theory. I have a number of comments that can I think be easily addressed.

My most substantial comment is that the results depend not just on the assumption of trade-offs per se, that are well underpinned empirically and theoretically, but also the shape of these trade-offs. Improved sensitivity comes at an accelerating costly lack of specificity and moreover this relationship is very strong and almost a threshold effect: considerable sensitivity is gained with relatively little loss of specificity up to a point when specificity falls off a cliff. These strong accelerating relationships are also assumed with the other fundamental trade-off such that you can crank up your immune response for initially little pathological cost but after a close to threshold point immunity gets really deadly. In both cases I am sure there are good mechanistic underpinnings for assuming these relationships, but I think they should be explained perhaps in detail in the figure legends. If there are limitations to our knowledge about the shape of these relationships, it would be good to

have these caveats here. Then around line 59 you could explain the importance of the shape of these relationships and any limitations. I think this would also help the reader with the intuitive explanation of the results.

More minor questions. Does the key result only occur with complex age structure? Would differential adult mortality in a two sex model with just juveniles and adults result in the same insights? Intuitively my sense was that it would, but I struggle with intuiting from continuous age distributions. Biologically I guess this may matter in stage as opposed to age structured systems (insects for example).

Are there any issues with survival being the currency of fitness here? This is knotty problem, but worth discussing although not too much. It makes perfect sense in the context of this problem since we particularly care about individual survival in the context of immune function.

In terms of presentation. I would rewrite the abstract to explicitly state that the two optima emerge naturally from the trade-offs, but that you need to understand why males and females are where they are. Clearly state that the classic explanation predicts the opposite? Then state that differential mortality due to parental care can explain --- at the moment it is a bit less specific and the message would benefit from the nice logic of the main paper.

Reviewer #4 (Remarks to the Author):

The authors present a series of models to understand the differing investment between males and females in pathogen detection (sensitivity) and strength of immune response. By combining two simple trade-offs for each arm of this dichotomy, they show a bimodal fitness landscape is generated. They then explore mechanisms that might drive the sexes to focus on opposite fitness peaks, suggesting that variable infection/mortality across lifespan and investment in parental care are the key processes.

This is an interesting study on an important and relevant question with novel findings. The methods look sound and are well described, and the manuscript is mostly well written. I did find a few passages confusing. This is for two main reasons. Firstly, I am not really a fan of theory papers where the methods come at the end, as it makes it quite hard to keep track of what is being done, and the reader has to jump back and forth continually. I guess this is an instruction from the journal, but if it is at all possible to move the methods (or at least some of the key results) earlier I think it would make the manuscript easier to read (for me at least). Secondly, some of the terminology I found a bit confusing, which I mention below.

Specific comments:

L57 - Is this really 'discrimination'? It is certainly 'sensitivity', but it strikes me that individuals are never actively 'discriminating' between what is self and non-self. They simply accept there is an overlap and judge where to draw the line. I'm also not clear on what is

meant by 'specificity' in this context? Why are highly sensitive immune systems not specific? Because they are more likely to pick up false positives?

L66 - In the methods you derive the result in term of s_p , but here in the text you refer to it in terms of s_e , which is a little confusing. It feels like the text focusses on sensitivity, so why not derive the result in terms of s_e ?

L100 - γ has not yet defined and is sort of buried in the methods. It might be helpful to plot fig 1A for a couple of different γ values to quickly see its impact. This might help with understanding figure 2 as well.

L130 - Please define λ either in the main text or on the axes in fig 3 E-H. (I know it's in the methods, but again, we've not seen this yet).

L160 - I have nothing against the conceptual argument per se, but I just wondered if it were possible to show this using your models? Or would adding in maternal transfer make it too unwieldy?

L205 - I don't immediately see why this is a probability, but it might just be me.

L212 - You should probably specify that this is the derivative wrt s_p .

Alex Best

Reviewer 2

Review for Metcalf and Graham

May 21, 2018

This manuscript presents a model that explores the trade-offs associated to immune detection to pathogens in immunological response (higher detection may also increase an autoimmune response) and with immune response, where higher response may also increase the damage to the host. The authors use this model to explain sex differences in immunological response from an optimization perspective. I find the premise and the aim of the paper fascinating; however, I have several doubts about the model and its implementation. I hope that my comments can help improve the presentation of the model. My main issue is that in general the formulation of the model seems to be determined by tractability more than reality. I think that justifying the functions that you chose in biological terms will greatly improve the credibility of the results. As it is now, it appears more as a mathematical exercise than as an attempt to explain actual biological processes.

- Lines 66-68: You dont define the subscript x for age while you dont provide the term for specificity (s_p).
- Do you envision that the hazards should also change with age x ?
- Line 99: When you provide the formula for R you use a term that you hadnt defined previously, μ_{id} . You could simply reference table 2 or provide an explanation.

1 Methods:

- From your formulation it seems that s_e and s_p are treated as probabilities, is this correct?
- Following up on the previous question, what is the range of values for s_p ? Based on your formula, it would seem that s_p should be bound between 0 and 1. However, if s_p is increasing between 0 and 1, the function for s_e would be monotonically declining with increasing values of s_p . To have it monotonically increasing as in Fig 1, you would need to have $s_e = 1 \exp(-\gamma s_p)$, where $s_p \geq 0$, or to have $s_p < 0$.

- About the function for survival s_x , is it the age-specific survival (i.e. $\Pr(X > x + \Delta x | X > x)$ for the random variable X for ages at death and observations $x \geq 0$) or the survivorship (i.e. $\Pr(X > x)$)? If its the latter, how did you define your hazards rate?
- It would seem to me that s_x is in fact the cumulative survival $\Pr(XZ > x)$, and therefore the exponent in the right hand side is equivalent to the cumulative hazards. Is that correct? Then what is the corresponding mortality function?
- If what you are modeling is in fact $\Pr(X > x + \Delta x | X > x)$, then you still have to define a hazards rate and a cumulative hazards since then $s_x = \exp \left[- \int_x^{x+\Delta x} \mu(t, \dots) dt \right]$, where $\mu(x, \dots)$ is the hazards rate or mortality function.
- Does age only enter the estimation of s_x through i_x ? This would imply that senescence and other age-related processes are only determined by the probability of infection at age x .
- Lines 213-219: you say that you take the derivative of s_x , the derivative with respect to what? Age x ?
- How is sex included in the model?
- Line 252: Provide reference for Leslie (1945).
- How do you choose the values for all your variables?

Response to reviewers

Reviewer #1 (Remarks to the Author):

This paper describes an extremely interesting model incorporating both sexual selection and demography in explaining sex differences in immune response. The idea of hazards and sensitivity is a very appropriate one to this question, and one that's quite novel. I liked the approach very much, but think that the generality of the work is overstated, and that it is applicable only to placental mammals, a point that needs to be made much more clearly. Furthermore, it isn't new to link life history and immunity. There is nothing wrong with these limitations, but they need to be emphasized from the start. The idea of parental immunity playing a key role is likewise new and interesting, though again I think it's limited to mammals and birds. I lack the mathematical expertise to evaluate the model itself, so leave that to other reviewers; I focus on the way that the results are interpreted and the conclusions are made.

Many thanks for these excellent points – we were perhaps a bit blinded by enthusiasm in the first draft. We have tried to be much more measured in this framing, making clear that much of the empirical evidence that we have used emerges from birds and mammals (with edits to the abstract and throughout the text), and that the novelty in the life history immunity link is actually more around in the combination of framing of immunity around sensitivity (removing the phrase “usually considered in isolation” from the abstract, and further edited the section in the closing paragraph. We also point to important directions for future research building off of the various suggestions and references you provide.

Starting with the title, the authors really don't mean “while parenting,” since this implies the difference is due to something happening during parental care, which most species don't exhibit. Similarly, later on, when they discuss “In the sex making a greater investment in parental care, mortality, infection risk, or mortality associated with infection, might all be increased relative to identically-aged individuals of the other sex,” they don't really mean parental care, they mean greater parental investment. Parental investment is seen in most if not all sexually-reproducing organisms, including plants, but parental care is only found in a tiny fraction. The distinction is important, because of course even in mammals, where females exhibit far more parental care than males, males still may have substantial parental investment in the form of, for example, expensive courtship displays. How this factors into the models is an interesting question, because parental investment by males may occur for a shorter but more intense period during the life history, say during mate attraction, but can still be quite high. So the contrast isn't between high and

low parental investment, it's the schedule of when those investments are made.

We appreciate this constructive and well-grounded critique. Our word choice needs to be better aligned with the core concepts we address. We've therefore revised to be much more careful about how we frame care vs. investment, and to clarify that the schedule of investments is the key component. Indeed, the (sensitivity-mediated) link between the schedule of investment and demographic outcomes is the central point here. Accordingly, the title of the manuscript is now: "Schedule and magnitude of reproductive investment under immune tradeoffs explains sex differences in immunity"

And where we introduce the points around age structure into the abstract, now stating that:

"By integrating these immune trade-offs into a life-history framework, where the schedule and magnitude of reproductive investment differs between the sexes,..."

And in the first sentence of the abstract: "There are clear sex differences in immunology in every species for which this has been investigated," again this seems like an over-generalization. Part of why the study of sex differences in immunity (not immunology, as the authors state) has been so complicated is that sometimes the sex differences are found and sometimes they are not – several reviews note the contradictions, particularly when non-mammals are included. I realize that the short format for Nature Communications requires brevity, but the authors need to be more careful about generalizing.

We've altered the first sentences of the abstract to capture this point: "Clear sex differences in immunity are found in many species. Probing the mechanistic underpinning of these differences in birds and mammals reveals greater nuance than the familiar caricature of males as the 'weaker' sex."

We've also tried to bring in many more of the nuances indicated by these comments and the references suggested (further detailed below).

Along similar lines, non-mammals in general and invertebrates in particular are very poorly studied in this field. This doesn't make the study less interesting, but it means that we simply don't know whether non-mammals exhibit similar trends. In particular, one study in pipefish doesn't make it the case that "Research on species for which sex roles are reversed (e.g., pipe-fish, where males carry offspring) is broadly consistent with this characterization." Are there studies of, say, phalaropes or spotted sandpipers or even marmosets (in which males provide a great deal of parental care) that substantiate this claim? Even calling sex roles "reversed" has been questioned, since it's not

clear that when a greater variety of species is taken into account, we can see clear roles for males and females across the board.

These are all excellent points, and we've opted to remove this sentence from the introduction. We have tried to end the manuscript by orienting towards this diversity of life histories and its potential for informing our premise, again with greater precision and recognition of the breadth of outcomes, both across species and within (via phenotypic plasticity), using the array of references suggested.

How is the idea that “qualitative immune differences between the sexes emerge from the interaction between trade-offs underpinning immune function and demographic differences between the sexes” different from other ideas that have been proposed? It looks to me like the demographic stuff is what's key here, since lots of other studies point to tradeoffs, even between immunity and other life history traits (see some of the references below).

We were struck by the emphasis on the magnitude of immune function in much previous work (e.g., among those you suggest, McKean & Nunnery falls very much along these lines) and felt that a strength of our approach is to explore predictions arising from 'qualitative' differences (detectors vs. effectors) instead of quantitative differences (in sheer magnitude of response) between the sexes. We've tried to make the sentence more reflective of the diversity of past work while retaining this sentiment:

“Here, we hypothesize that qualitative immune differences between the sexes (beyond the quantitative differences in immune response magnitude that are often a focus, e.g. ¹⁴) emerge from the interaction between trade-offs underpinning immune function and demographic differences between the sexes.”

The idea that testosterone is or isn't key to the tradeoff is interesting, since of course (related to the above points) most animals on earth don't have it. If invertebrates show the same pattern of generally higher female immunity, what would this say about how this pattern evolved? It seems to me that one can argue that testosterone's effect on the immune system may be an incidental mechanism, but that differential selection on males and females would produce sex differences in immunity regardless. Alternatively, testosterone's influence could be a necessary precursor for the kind of sex differences we see, because in animals that lack such hormones, no other pathways make the link between immunity and sexual selection possible. We just don't know, though a number of investigators have tried to examine sex differences in immunity in invertebrates. The bottom line, though, is whether this paper and its findings are just about mammals or perhaps vertebrates, and it's hard to see from the manuscript what the authors think.

Given your excellent points, we've revised to clarify our focus on vertebrates and the need for more data on invertebrates. We address this in the discussion:

“Strengthening the empirical knowledge-base of sex differences across a broader array of species across the tree of life will be essential to evaluating emergent predictions. For example, although immune defenses of invertebrates, including immune memory and maternal immunity (e.g., ²⁹⁻³¹), are increasingly well understood, proximate drivers analogous to likely mediators of sex effects in birds and mammals (such as testosterone ²²) remain largely unknown. Furthermore, elucidation of plasticity in sex differences ¹⁴ and the role of genetic constraints on sexual dimorphism on shaping life history outcomes ³² are important directions both theoretically and empirically.”

A strength of the paper is its look at different life history stages, not just assuming that all adults have the same tradeoffs between sensitivity and other traits. I also think that the literature on tolerance vs. resistance is relevant to the authors' argument, because the idea is the same as their sensitivity/false positive vs. false negative approach; there, the tradeoff is between ramping up one's immunity to resist a pathogen, which is costly, or not do so and simply deal with (tolerate) the effects, which may be less costly in the long run. Lars Raberg has done some of this work.

Potential links with the tolerance/resistance literature are fascinating. We've thought about this, and we find that formally making the connection to our framework would be somewhat intricate, as follows. Tolerance is usually framed as the ability to sustain health despite high/increasing pathogen load, while resistance is framed as the ability to prevent high pathogen loads. Putting this in the terms of our framework, an increase in tolerance should reflect a reduction in the hazard associated with infection μ_d , with no additional cost to health (implying no additional immunopathology). However, importantly, some form of costs is of course necessary for this to be evolutionarily tenable (otherwise no variation in the trait is expected). Likewise, an increase in sensitivity could be thought of as limiting pathogen loads, and thus increasing resistance – however, again, the question of what to do with costs of this increase generally, and how and where immunopathology costs enter is again a question – in our framing, immunopathology costs are inevitable, but this is not the case for more general tolerance/resistance models.

To frame this another way, the two peaks that emerge on Figure 1C do not just recapitulate 'resistant' versus 'tolerant' strategies, because each peak arguably encompasses a bit of each. One could summarize our results as showing that females resist with great detection but tolerate with moderate magnitude, while males tolerate with poor detection but resist with high magnitude responses. We've pointed to this distinction in the manuscript, without going to further into the details in order to avoid detracting from our main points:

“Finally, we note that tradeoffs surrounding immune sensitivity and response magnitude are likely to shape whether a host resists and/or tolerates pathogens²⁰ but that the two survival peaks described above do not simply equate to resistance and tolerance. ”

Overall, while we do think this is a really interesting area for further extension, we feel that given the array of potential directions for exploring this (particularly the potential complex physiological trade-offs that could underlie the mortality hazard tradeoffs), this is beyond the scope of what we present here.

I understand that the authors are limited in their citations, but here are a few very relevant papers that would be of interest:

Zuk, M. and K.A. McKean. 1996. Sex differences in parasite infections: patterns and processes. *Int. J. Parasitol.* 26:1009-1024.

Rolff J, Armitage SAO & Coltman DW (2005) Genetic constraints, sexual dimorphism and immune defence. *Evolution*, 59, 1844-1850.

Rolff J (2002) Bateman's principle and immunity. *Proceedings of the Royal Society of London B*, 269, 867-872.

McKean, K.A. and Nunney, L. 2005. Bateman's principle and immunity: phenotypically plastic reproductive strategies predict changes in immunological sex differences. *Evolution* 59(7): 1510-1517.

Moore, S.L. and Wilson, K. 2002. Parasites as a viability cost of sexual selection in natural populations of mammals. *Science* 297: 2015-2018.

Zuk, M. and Stoehr, A.M. 2002. Immune defense and life history. *The American Naturalist* 160: s9-s22.

Pursall, E.R. and Rolff, J. 2012. Immunopathology in ecological immunology. In: *Ecoimmunology*, G.E. Demas and R.J. Nelson, eds. Oxford University Press.

Many thanks for these pointers - we have included a number of these references throughout the text, but especially in the closing sentences of the manuscript (as mentioned above).

Reviewer #2 (Remarks to the Author):

Please see attached file.

Review for Metcalf and Graham

May 21, 2018

This manuscript presents a model that explores the trade-offs associated to immune detection to pathogens in immunological response (higher detection may also increase an autoimmune response) and with immune response, where higher response may also increase the damage to the host. The authors use this model to explain sex differences in immunological response from an optimization perspective. I find the premise and the aim of the paper fascinating; however, I have several doubts about the model and its implementation. I hope that my comments can help improve the presentation of the model. My main issue is that in general the formulation of the model seems to be determined by tractability more than reality. I think that justifying the functions that you chose in biological terms will greatly improve the credibility of the results. As it is now, it appears more as a mathematical exercise than as an attempt to explain actual biological processes.

- Lines 66-68: You dont define the subscript x for age while you dont provide the term for specificity (s_p).
- Do you envision that the hazards should also change with age x ?
- Line 99: When you provide the formula for R you use a term that you hadnt defined previously, μ_{id} . You could simply reference table 2 or provide an explanation.

1 Methods:

- From your formulation it seems that s_e and s_p are treated as probabilities, is this correct?
- Following up on the previous question, what is the range of values for s_p ? Based on your formula, it would seem that s_p should be bound between 0 and 1. However, if s_p is increasing between 0 and 1, the function for s_e would be monotonically declining with increasing values of s_p . To have it monotonically increasing as in Fig 1, you would need to have $s_e = 1 \exp(-\gamma s_p)$, where $s_p \geq 0$, or to have $s_p < 0$.

- About the function for survival s_x , is it the age-specific survival (i.e. $\Pr(X > x + \Delta x | X > x)$) for the random variable X for ages at death and observations $x \geq 0$) or the survivorship (i.e. $\Pr(X > x)$)? If its the latter, how did you define your hazards rate?
- It would seem to me that s_x is in fact the cumulative survival $\Pr(XZ > x)$, and therefore the exponent in the right hand side is equivalent to the cumulative hazards. Is that correct? Then what is the corresponding mortality function?
- If what you are modeling is in fact $\Pr(X > x + \Delta x | X > x)$, then you still have to define a hazards rate and a cumulative hazards since then $s_x = \exp \left[- \int_x^{x+\Delta x} \mu(t, \dots) dt \right]$, where $\mu(x, \dots)$ is the hazards rate or mortality function.
- Does age only enter the estimation of s_x through i_x ? This would imply that senescence and other age-related processes are only determined by the probability of infection at age x .
- Lines 213-219: you say that you take the derivative of s_x , the derivative with respect to what? Age x ?
- How is sex included in the model?
- Line 252: Provide reference for Leslie (1945).
- How do you choose the values for all your variables?

Many thanks for the comments, which we have now addressed and hope have improved the presentation of the model. As you rightly point out, tractability was an important motivation in the model development – but we do think that, although our framing is inevitably a caricature of immune system function and the broader life history context, the general insights that emerge from this simple framing do justify our approach. It is also clear both from your comments, and from those from reviewer 4, that the way in which we introduced the methods was rather confusing. We’ve substantially revisited this and tried to be much clearer.

L66 – we have improved introduction of core terms around this point, in line also with suggestions from reviewer 4.

Hazards are of course very likely to change with age (an important foundation of much interesting work in demography). In this first investigation, we have kept things as simple as possible. We now explicitly acknowledge this in our introduction of the methods: “At every age, individuals experience a background hazard of mortality μ_b (assumed constant over age, for simplicity) which may be increased....”

And, below, where modulation of the background hazards are used as short-hand for the outcome of other trade-offs and investments (with, e.g., lower survival during reproductive years) we have tried to make this very clear: “Taking as a starting point constant background mortality over age, and constant fertility from the age of maturity, we can characterize the consequences of a reduction in survival during reproductive years (achieved by increasing the background hazard of mortality μ_b during those years), potentially reflecting costs associated with parental investment (Figure 3A)...”

However, we do feel that one very interesting direction for development would be to start titrating the implications of age specific differences in background hazards for other important life history questions, such as, e.g., differences between the sexes in the evolution of senescence. We mention the importance of exploring varying age trajectories in future work in the final paragraph: “Encompassing realistic age-schedules of survival (including high early and late mortality) and fertility (beyond the constant rates considered here)...”

L99 – apologies for missing this, we now introduce the term

Methods:

We are indeed treating s_e and s_p as probabilities, in line with the basic framing from epidemiology, and indeed, the expression defined is a monotonically declining function (as expected, since this inherently reflects a trade-off that we are harnessing to explore evolutionary outcomes). This negative relationship does not match the lower panel of e.g., Figure 1A, because the tradition from epidemiology is to plot the former (s_e) against one minus the latter ($1-s_p$). We’ve revised the caption to Figure 1 to highlight this, and also to make clear that these values are both bounded between 0 and 1: “Host immunity then faces a trade-off between reducing false positives (triggering a response in the absence of pathogens) and reducing false negatives (failing to detect pathogens), as delineated by a classic Receiver Operator Curve (ROC) from epidemiology obtained by plotting sensitivity (proportion of true positives detected) against 1-specificity (proportion of true negatives detected).” We’ve also

considerably expanded the caption to Figure S1 to make this as transparent as possible.

In our framing, s_x is indeed age specific survival, rather than survivorship; many thanks for pushing us to make this clarification, which is now made explicit in the text. Within the methods, where we introduce use of the Leslie matrix, we clarify that we will thus be in a situation where the product of survival at each age (and thus the sum of the underlying hazards) is required to estimate survivorship to every age: “We assume that survival from age x to $x+1$, reflecting the x^{th} column and $x+1^{\text{th}}$ row in the matrix, is defined by s_x (such that the proportion of individuals surviving to age A will be $s_1 s_2 s_3 \dots \times s_{A-1}$)”

We also realize that we had originally introduced use of a full demographic model and our proxies for fitness rather before it was actually used (i.e., around the optimality modelling of R and γ) – accordingly, we have shifted it down to where it more appropriately belongs, i.e., at the start of the text relating to Figure 3.

“In the sex making a greater parental investment, their mortality, infection risk, or mortality associated with infection, might all be increased relative to identically-aged individuals of the other sex. Such changes will shift both the stable population structure and reproductive value of individuals across ages, altering the impact of underlying parameters such as immune sensitivity on the population growth rate, a proxy for fitness (see Methods). Addressing this requires moving beyond identifying the strategy that optimizes, s_x , survival within an age class (the focus so far) to identifying the strategy that maximizes fitness.”

It is indeed the case that, in the current formulation, only infection at age x is modulating the age profile of survival. There are many fascinating questions about the evolution of senescence that we have not dug into here, but indicate great directions future work, now mentioned in the discussion, as indicated above.

L213 – apologies, we now specify what the derivative is relative to (s_p)

The assumptions around how we modelled sex were buried in the methods - we’ve pulled them out to be more central in this version. We now state at the first mention of the demographic framing:

“Addressing this requires moving beyond identifying the strategy that optimizes, s_x , survival within an age class (the focus so far) to identifying the strategy that maximizes fitness for each of the sexes, i.e., requiring a full demographic model. We assume that neither sex is limiting, and therefore evaluate outcomes for the two sexes separately (see Methods).”

And, in the methods, we introduce the ways in which this fails to account for a limiting sex, and how this might be addressed:

“For comparison between the sexes, this is equivalent to making the assumption that the mate of the opposite sex is never limiting (i.e., we are not formally modeling a ‘marriage function’).”

In the final statements for future directions we bring in the importance of formally accounting for the degree to which one of the two sexes may be limiting:

“...as well as more formally accounting for the degree to which one of the two sexes may be limiting (e.g., by including a ‘marriage function’³⁴) may generate more nuanced system-specific predictions.”

We’ve included the reference for Leslie 1945, many thanks.

Our choice of values for variables was driven by a combination of natural constraints (e.g., s_e and s_p are constrained between 0 and 1) and ranges where interesting results might emerge – for example, ranges with very high mortality associated with infectious disease could never reveal a benefit for lower sensitivity. Ideally, such values would be mapped onto empirical evidence, but the paucity of data on causes of death, combined with the difficulty of quantifying competing hazards even where such data are available, means that this is a rich area for future developments.

Reviewer #3 (Remarks to the Author):

Metcalf and Graham examine theoretically the important question of why males and females typically differ in immune function. A real advance here is the mechanistic detail at which the authors address this question. Typically, previous studies have been much more superficial when addressing this question with a simple characterization that males have reduced immune function due to immunosuppression through testosterone which is used as a hypothesis to explain male biased parasitism alongside higher exposure in males due to behavioral differences. What is very novel here is that it is underpinned by a much more detailed understanding of sex differences in immune function: with females better at detection and males better at the response. Cutting edge life-history theory is used to determine how we can understand these differences from first principles. Well empirically underpinned trade-offs are assumed that convincingly show that these bimodal life-histories can be easily predicted to coexist. Then they examine what factors can lead to males and females being selected for less often stronger and more common weaker responses respectively. What I really like is that they show that the classic Bateman’s principle arguments do not work, but that the pattern can be explained by differential mortality due to parental care. This gives testable predictions and changes the way the field thinks about sex differences in immune function. I would expect it to lead to shift in the field and be exceptionally well cited.

Many thanks for the wonderful encouragement.

The paper is very well written and authoritative. It clearly benefits from this unique collaboration between two evolutionary biologists with complimentary expertise in immunology and life-history/demographic theory. I have a number of comments that can I think be easily addressed.

My most substantial comment is that the results depend not just on the assumption of trade-offs per se, that are well underpinned empirically and theoretically, but also the shape of these trade-offs. Improved sensitivity comes at an accelerating costly lack of specificity and moreover this relationship is very strong and almost a threshold effect: considerable sensitivity is gained with relatively little loss of specificity up to a point when specificity falls off a cliff. These strong accelerating relationships are also assumed with the other fundamental trade-off such that you can crank up your immune response for initially little pathological cost but after a close to threshold point immunity gets really deadly. In both cases I am sure there are good mechanistic underpinnings for assuming these relationships, but I think they should be explained perhaps in detail in the figure legends. If there are limitations to our knowledge about the shape of these relationships, it would be good to have these caveats here. Then around line 59 you could explain the importance of the shape of these relationships and any limitations. I think this would also help the reader with the intuitive explanation of the results.

This is an excellent point, and we've now tried to be much more detailed in our framing around Figure 1A and B.

First, in the caption of Figure S1, we more completely describe the determinants of the shape of the relationship linking sensitivity and specificity. We also point to the potential of modified trade-off shapes altering the optimal landscapes. Second, we clarify what we do and don't assume about the shape of the relationship between immunopathology and defence. While the trade-off is depicted as falling off exponentially, in Figure 1B, and we do use an exponential framing for convenience to pin down dependencies of the optima through our analyses of the derivatives, in actual fact the shape doesn't enter into subsequent analyses. We state this formally in the text:

“The shape of two trade-offs (the first, linking sensitivity and specificity, and the second, reflecting positive and negative effects of immune effector response) will also influence these outcomes¹⁹. The former is to some degree constrained (illustrated and discussed in Figure S1), and while the exponential framing used above for the latter simplifies derivation of the optimal magnitude of the immune effector response (above), the bimodal survival landscape emerges for a range of shapes of the relationship linking μ_i and μ_d (see code for an example with a linear relationship). Additionally, the remainder of results presented do not depend on the exact shape of the relationship (beyond it being negative).”

And the code submitted alongside the manuscript now includes a function called ‘testShapeTradeOff’ which produces Figure 1C for a linear rather than exponential trade-off. In Table 1, where we introduce the analytical form for the trade-offs around immune effectors, we also make clear that this is just one option; and in the caption to Figure 1, we “note that a variety of trade-off shapes are possible”.

Finally, in the methods, we state that:

“Note that the two trade-off relationships defined above for μ_i and μ_{id} as a function of μ_d are only used to obtain analytical results for the optimal investment in immune effectors, and do not influence the broader set of qualitative results, see text.”

More minor questions. Does the key result only occur with complex age structure? Would differential adult mortality in a two sex model with just juveniles and adults result in the same insights? Intuitively my sense was that it would, but I struggle with intuiting from continuous age distributions. Biologically I guess this may matter in stage as opposed to age structured systems (insects for example).

You’re absolutely right that the nuances of age-structure are not really explored in detail here – differential adult mortality would suffice to recapture most of our results. The essential issue is one of increased mortality at earlier ages increasing selection for survival at later ages, all else being equal (introduced in Abrams, P.A. (1993) Does increased mortality favor the evolution of more rapid senescence? *Evolution* 47 (3), 877-887.; Williams, P.D. and Day, T. (2003) Antagonistic pleiotropy, mortality source interactions, and the evolutionary theory of senescence. *Evolution* 57 (7), 1478- 1488.). However, we felt that illustrating the scope of declining selection pressures across a range of ages (as shown in Figure 3) and particularly the opportunity that this provided for exploring the impact of increasing fertility at late ages (Figure 3C, orange line) helped the intuition underpinning our analysis, and provided us one simple line for thinking about sex differences in fertility. We now address the added layers of nuance that more detailed age structure might bring in, in response also to other reviewers:

“Encompassing realistic age-schedules of survival (including high early and late mortality) and fertility (beyond the constant rates considered here) for which there is increasing information²⁹, as well as more formally accounting for the degree to which one of the two sexes may be limiting (e.g., by including a ‘marriage function’³⁰) may generate more nuanced system-specific predictions.”

Are there any issues with survival being the currency of fitness here? This is knotty problem, but worth discussing although not too much. It makes perfect sense in the context of this problem since we particularly care about individual survival in the context of immune function.

While survival differences entirely define the optimal landscapes in Figure 1 and interpretation underpinning Figure 2 (and we've tried to be careful not to use the term fitness around these results), subsequent analyses are built around a formal demographic model, and use the population rate of increase as our proxy of fitness. The focus is still very survival-centric – and of course sex-differences in schedules of fertility could importantly alter the optimal level of sensitivity. We hint at this in our analysis of the impact of increasing fertility at late ages (Figure 3C, orange line, and Figure S4), but have also tried to bring this out more in the discussion alongside a discussion of the fitness measure used as requested by another reviewer; see paragraph quoted on previous point.

In terms of presentation. I would rewrite the abstract to explicitly state that the two optima emerge naturally from the trade-offs, but that you need to understand why males and females are where they are. Clearly state that the classic explanation predicts the opposite? Then state that differential mortality due to parental care can explain --- at the moment it is a bit less specific and the message would benefit from the nice logic of the main paper.

This is a really wonderful suggestion, and we've tried to reframe the abstract along the lines suggested. Many thanks!

Reviewer #4 (Remarks to the Author):

The authors present a series of models to understand the differing investment between males and females in pathogen detection (sensitivity) and strength of immune response. By combining two simple trade-offs for each arm of this dichotomy, they show a bimodal fitness landscape is generated. They then explore mechanisms that might drive the sexes to focus on opposite fitness peaks, suggesting that variable infection/mortality across lifespan and investment in parental care are the key processes.

This is an interesting study on an important and relevant question with novel findings. The methods look sound and are well described, and the manuscript is mostly well written. I did find a few passages confusing. This is for two main reasons. Firstly, I am not really a fan of theory papers where the methods come at the end, as it makes it quite hard to keep track of what is being done, and the reader has to jump back and forth continually. I guess this is an instruction from the journal, but if it is at all possible to move the methods (or at least some of the key results) earlier I think it would make the manuscript easier to read (for me at least). Secondly, some of the terminology I found a bit confusing, which I mention below.

We agree that the paper is a bit cryptic as a result of the way the methods were embedded. To make it easier to read, we now introduce the methods a bit more, at

an earlier stage, and are much more explicit about our methods at each extension of the analysis. We also realized that we had somewhat confusingly introduced methods relating to the demographic model in the text rather before they were used, and so we have changed this around.

Specific comments:

L57 - Is this really ‘discrimination’? It is certainly ‘sensitivity’, but it strikes me that individuals are never actively ‘discriminating’ between what is self and non-self. They simply accept there is an overlap and judge where to draw the line. I’m also not clear on what is meant by ‘specificity’ in this context? Why are highly sensitive immune systems not specific? Because they are more likely to pick up false positives?

We’ve retained the use of the term ‘discrimination’ from the epidemiology literature in part because we feel that it makes discussion of the gradient between weak to strong ability to separate self and non-self (e.g., illustrated under the x axis in Figure 2) easier to discuss. The trade-off between sensitivity and specificity (true positives come at the expense of false positives, which reflect low specificity) is a linchpin of the work and hopefully should be more transparent in this version: following suggestions from the other referees, we’ve extended our discussion and definition of specificity and its tight reliance on sensitivity, in the caption to Figure 1, Figure S1, and in Table 1.

L66 - In the methods you derive the result in term of s_p , but here in the text you refer to it in terms of s_e , which is a little confusing. It feels like the text focusses on sensitivity, so why not derive the result in terms of s_e ?

This is somewhat a legacy issue, to echo the framing we used in a previous paper (Metcalf, Tate & Graham, 2017, Nature Ecology and Evolution) to allow easy comparison. However, for the main text, we felt that sensitivity was the more intuitive metric, and thus used this as the core framing device in all explanations for consistency. We now note this in the text of the methods (“And this is straightforward to relate to the optimal sensitivity, s_e^* , results for which are discussed in the main text (analytical results are framed in terms of specificity for consistency with previous work¹⁵)” and also mention this point in the code that accompanies the submission.

L100 - gamma has not yet defined and is sort of buried in the methods. It might be helpful to plot fig 1A for a couple of different gamma values to quickly see its impact. This might help with understanding figure 2 as well.

We tried a few options for showing different values of gamma on Figure 1, but they all came out as rather more confusing. We’ve taken the path of considerably expanding the caption to Figure S1. We have also expanded the text around this first mention of gamma:

“Males which reduce investment in the magnitude of the immune response (e.g., reducing $R = \frac{\mu_i}{\mu_d - \mu_{id}}$, thus shifting from the top left to bottom right on Figure 1B, see Table 2 for term definitions, and noting that the shape of the relationship linking the various hazards does not enter into this expression) or in discrimination (e.g., reducing γ , the parameter governing the shape of the relationship between sensitivity and specificity, and thus increasing the perceived overlap between the distribution of self and non-self, Figure 2, lower panel, Figure S1), might experience a fertility advantage.”

L130 - Please define lambda either in the main text or on the axes in fig 3 E-H. (I know it's in the methods, but again, we've not seen this yet).

Many apologies - we've now introduced the term in the text at first mention, and edited the figure caption to introduce this: “*Effect of changes of survival on fitness* as measured by the populations rate of increase, λ , is the product of the stable population structure (B) and reproductive value”

L160 - I have nothing against the conceptual argument per se, but I just wondered if it were possible to show this using your models? Or would adding in maternal transfer make it too unwieldy?

We thought hard about how we could do this, but since it is necessary to keep track of the stochastic array of pathogens that mothers have experienced through time to transfer in order to transfer this information to their offspring, we couldn't think of a very tractable approach. The conceptual framing underpinning Figure 4 (although broadly correct) neglects the issue of the changing and non-uniform distributions of self and non-self. To ensure that our conclusions were robust to these conclusions, we also developed an Individual Based Model, for which the code is also submitted alongside this manuscript. We've tried to introduce the individual based model more clearly in this version:

“Under these conditions, sensitivity in females (or more generally the immunity-transferring sex) will increase (see Text S1 for details of a simulation to support the conceptual framing, accompanying code, and Figure S5 for maternal immunity results, Figure S6 -S7 for determinants of the optima).”

L205 - I don't immediately see why this is a probability, but it might just be me.

This could be framed a few different ways (and interpreted a few different ways). We found the “probability of being infected” the most intuitive, so have retained that framing here.

L212 - You should probably specify that this is the derivative wrt s_p .

Done, many thanks.

Alex Best

Reviewers' comments:

Reviewer #1 (Remarks to the Author):

Thanks for your thoughtful responses to the reviewers' comments -- I have no further suggestions and think this is a very valuable and interesting contribution.

Reviewer #2 (Remarks to the Author):

see attached file.

Reviewer #3 (Remarks to the Author):

I like the paper before and I think it is now much improved. Their replies to referee have put the has allowed a better focus on the degree of generality of the results. Being upfront that the evidence is missing beyond mammals strengthens the impact of the paper. The model nicely balances realism and tractability in my view. I think it is an original contribution highlighting the key differences between males and females in terms of detection, showing that these peaks can emerge from a simple trade off and determine which processes would send males and females to these peaks. The dependency of the results on assumptions of the shape of trade-offs is much clearer now and it is encouraging that this is less important than I had thought. The paper has a clearer, more logical structure and is now likely to make a big impact. The key message that these patterns can be explained by shared immune trade-offs but sex specific reproductive schedules and risks. This should change the way sex biased immune responses are understood in the literature. I think it should be accepted as is.

Small suggestions

Line 50 – no need for the e.g. with a reference – it might be just me but it looks horrible.

Line 103 – would it be better to say “may combine both resistance and tolerance” here?

Reviewer #4 (Remarks to the Author):

This resubmission is a good improvement on the previous version, which I was already pretty positive about. The authors have either made relevant changes based on my comments or given good reasons to not do so. More broadly, many of the changes make the manuscript rather easier to read and less confusing.

My final comment/request is that you include the parameter values used to produce your plots in the figure legends. I have been trying to reproduce figure 1C in particular (as I was

interested by the comment that the behaviour stays the same even for a linear trade-off). I have been using the values that seem to be in your R code (I don't use R myself so have been doing it in Maple) and don't seem able to reproduce the plot you have in 1C.

Reviewer 2

Second review for Metcalf and Graham

July 16, 2018

I really enjoyed reading this new version and believe that the results are extremely interesting and will have important implications. I believe that you have answered most of my comments, I have only some minor comments on this new version.

- Paragraph starting in line 83: This is certainly one of the key results, however, referring to Fig 1C you state that the results suggest that low sensitivities are associated with high immune response and vice versa. However, as I understand your model, it's not the immune response per se that you are modeling, but the hazard associated to it, both in the absence of infection and in the presence of it. It sounds logical to assume that the hazards will be positively related to the immune response, but it seems to me that you are assuming a proportional relationship between the two, is this correct?
- Based on the equation for s_x in line 241, the signs in the equation for s_x are inverted.
- Also, you can verify that it is a maximum if $d^2s_x/dx^2 < 0$ (i.e. the function is concave at s_p^*). You should also verify the equation for the optimal specificity, I believe that in the second logarithm on the right-hand-side you should have $i_x - 1$ in the numerator.
- The signs in the equation for $ds_x/d\mu_x$ are also inverted. Maybe the problem is in the original equation for s_x . Actually, I am convinced that the problem is in the equation for s_x . If you define your hazard for age x as

$$\mu(x) = \mu_b + (1 - i_x)\mu_i(1 - s_p) + i_x\mu_d e^{-\gamma(1-s_p)} + i_x\mu_i d \left[1 - e^{-\gamma(1-s_p)} \right],$$

with cumulative hazards approximated as

$$H(x) = \sum_{t=0}^x \left\{ \mu_b + (1 - i_t)\mu_i(1 - s_p) + i_t\mu_d e^{-\gamma(1-s_p)} + i_t\mu_i d \left[1 - e^{-\gamma(1-s_p)} \right] \right\}$$

with survivorship $S(x) = e^{-H(x)}$. The age-specific survival then is

$$\begin{aligned} s_x &= \frac{S(x+1)}{S(x)} \\ &= \exp \left\{ -\mu_b - (1 - i_{x+1})\mu_i(1 - s_p) - i_{x+1}e^{-\gamma(1-s_p)} - i_{x+1}\mu_{id} \left[1 - e^{-\gamma(1-s_p)} \right] \right\}. \end{aligned}$$

- In terms of notation, if you have s_x then you expect the derivative to be with respect to x . I suggest that, to avoid confusions, you use the notation ds_x/ds_p and $ds_x/d\mu_d$.
- Fig 2: shouldnt you have $R = \mu_i/(\mu_d - \mu_{id})$? i.e. the sign in the denominator.

Reviewers' comments:

Reviewer #1 (Remarks to the Author):

Thanks for your thoughtful responses to the reviewers' comments -- I have no further suggestions and think this is a very valuable and interesting contribution.

Many thanks!

Reviewer #2 (Remarks to the Author):

I really enjoyed reading this new version and believe that the results are extremely interesting and will have important implications. I believe that you have answered most of my comments, I have only some minor comments on this new version.

Thank you so much for your positivity – and thanks so much for the very careful review!

Paragraph starting in line 83: This is certainly one of the key results, however, referring to Fig 1C you state that the results suggest that low sensitivities are associated with high immune response and vice versa. However, as I understand your model, it's not the immune response per se that you are modeling, but the hazard associated to it, both in the absence of infection and in the presence of it. It sounds logical to assume that the hazards will be positively related to the immune response, but it seems to me that you are assuming a proportional relationship between the two, is this correct?

I think the confusion here emerges from the way we've tried to explain two different facets of immunity on one composite figure. Indeed, it is correct that the hazard emerges from the immune response – but it emerges from two aspects of the immune response: both the 'sensitivity / specificity' aspects (Figure 1A) and 'magnitude of response' aspects (Figure 1B). We've edited this paragraph to try and make this more clear, by harking back specifically to Figure 1B where appropriate, e.g.:

“To maximize survival, strategies with high sensitivity are paired with a low magnitude immune response, i.e., one involving few effector cells (Figure 1C, top right) and vice versa (Figure 1C, bottom left). Where sensitivity is high, the mortality hazards associated with infection are experienced rarely, so the survival benefits of a reduced magnitude immune response associated with reduced immunopathology (reflected by points towards the right of the trade-off on Figure 1B) outweigh the costs of reduced response to infection (and thus high mortality hazard associated with infection, x axis, Figure 1B).”

Based on the equation for s_x in line 241, the signs in the equation for s_x are inverted.

This is terribly embarrassing, and of course, you're absolutely right, there is a bracket missing in the equation. Thank you so much for taking the time to identify it. The expression

is correct in the code, and was implemented correctly in the analysis; we have altered the equation in the methods to the correct formulation.

Also, you can verify that it is a maximum if $d^2s_x/dx^2 < 0$ (i.e. the function is concave at s_p^*). You should also verify the equation for the optimal specificity, I believe that in the second logarithm on the right-hand-side you should have $i_x - 1$ in the numerator.

We worked through to verify that the derivative of survival relative to s_p (but this time correctly framed! thank you) does indeed emerge as:

$$\frac{ds_x}{ds_p} = s_x [(1 - i_x)\mu_i - \gamma i_x \mu_d \exp^{-\gamma(1-s_p)} + \gamma i_x \mu_{id} \exp^{-\gamma(1-s_p)}]$$

(following the notation change that you also suggest). From this, setting the right-hand component to zero, we have:

$$(1 - i_x)\mu_i + \gamma i_x \exp^{-\gamma(1-s_p)}(\mu_{id} - \mu_d) = 0$$

$$(1 - i_x)\mu_i = \gamma i_x \exp^{-\gamma(1-s_p)}(\mu_d - \mu_{id})$$

$$\frac{(1 - i_x)\mu_i}{\mu_d - \mu_{id}} \frac{1}{\gamma i_x} = \exp^{-\gamma(1-s_p)}$$

$$\log\left(\frac{[1 - i_x]}{i_x}\right) + \log\left(\frac{\mu_i}{\mu_i - \mu_{id}}\right) - \log(\gamma) = -\gamma(1 - s_p)$$

From which,

$$s_p = \frac{1}{\gamma} \left[\log\left(\frac{[1 - i_x]}{i_x}\right) + \log\left(\frac{\mu_i}{\mu_i - \mu_{id}}\right) - \log(\gamma) \right] + 1$$

which matches the optimal reported in the paper (the previous typo that you caught presumably obscured this).

As you suggest, we also evaluated the second derivative of survival at the optimal. For convenience, we define $A = (1 - i_x)\mu_i + \gamma i_x \exp^{-\gamma(1-s_p)}(\mu_{id} - \mu_d)$. Then, from the definition of survival, we have:

$$\frac{ds_x}{ds_p} = s_x A,$$

from which the second derivative of survival at the optimal is:

$$\frac{d^2s_x}{d^2s_p} = s_x \left[A^2 + \frac{dA}{ds_p} \right] \quad \text{and} \quad \frac{dA}{ds_p} = \gamma^2 i_x \exp\left(-\gamma(1 - s_p)\right) [\mu_{id} - \mu_d].$$

At the optimal for s_p defined above, we have:

$$\exp(-\gamma(1 - s_p)) = \frac{(1-i_x)}{i_x} \frac{\mu_i}{\mu_i - \mu_{id}} \frac{1}{\gamma},$$

and introducing this into the expression for $\frac{dA}{ds_p}$, yields:

$$\frac{dA}{ds_p} = \gamma(1 - i_x)(-\mu_i)$$

which is negative, as γ and μ_i are defined as positive, and i_x is bounded by 0 and 1. Since A^2 is zero at the optimal, and s_x is positive, we can conclude that the second derivative is negative. We now mention this at the relevant point in the methods:

“Further, introducing s_p^* into the expression for the second derivatives of survival at the optimal, $\frac{d^2s_x}{d^2s_p}$ yields a negative value, confirming that this value reflects a maximum.”

And report the same results obtained for μ_d^* following the same logic:

“which can be directly related to the optimal hazard associated with immunopathology, μ_i^* , which scales with the magnitude of the immune response deployed (see main text); again, second derivatives are negative indicating that this corresponds to a maximum.”

The signs in the equation for $ds_x/d\mu_x$ are also inverted. Maybe the problem is in the original equation for s_x . Actually, I am convinced that the problem in the equation for s_x .

As mentioned above, you are indeed right, and thank you so much for catching this. The signs are correct further down – the original error was the source of the confusion.

In terms of notation, if you have s_x then you expect the derivative to be with respect to x . I suggest that, to avoid confusions, you use the notation ds_x/ds_p and $ds_x/d\mu_d$.

We have changed this notation as suggested.

Fig 2: shouldn't you have $R = \mu_i/(\mu_d - \mu_{id})$? i.e. the sign in the denominator.

Oh gosh – another embarrassing mistake – we've altered this also as suggested (and it is correct in both the text and the analyses). Thanks so much again.

Reviewer #3 (Remarks to the Author):

I like the paper before and I think it is now much improved. Their replies to referee have put the has allowed a better focus on the degree of generality of the results. Being upfront that the evidence is missing beyond mammals strengthens the impact of the paper. The model nicely balances realism and tractability in my view. I think it is an original contribution highlighting the key differences between males and females in

terms of detection, showing that these peaks can emerge from a simple trade off and determine which processes would send males and females to these peaks. The dependency of the results on assumptions of the shape of trade-offs is much clearer now and it is encouraging that this is less important than I had thought. The paper has a clearer, more logical structure and is now likely to make a big impact. The key message that these patterns can be explained by shared immune trade-offs but sex specific reproductive schedules and risks. This should change the way sex biased immune responses are understood in the literature. I think it should be accepted as is.

Many thanks!

Small suggestions

Line 50 – no need for the e.g. with a reference – it might be just me but it looks horrible.

In retrospect, it does rather... have edited. Many thanks.

Line 103 – would it be better to say “may combine both resistance and tolerance” here?

Edited as suggested.

Reviewer #4 (Remarks to the Author):

This resubmission is a good improvement on the previous version, which I was already pretty positive about. The authors have either made relevant changes based on my comments or given good reasons to not do so. More broadly, many of the changes make the manuscript rather easier to read and less confusing.

Many thanks!

My final comment/request is that you include the parameter values used to produce your plots in the figure legends. I have been trying to reproduce figure 1C in particular (as I was interested by the comment that the behaviour stays the same even for a linear trade-off). I have been using the values that seem to be in your R code (I don't use R myself so have been doing it in Maple) and don't seem able to reproduce the plot you have in 1C.

We added the parameters to the caption of Figure 1C – hope it works now!